# Temporal Regularized Matrix Factorization for High-dimensional Time Series Prediction

**Hsiang-Fu Yu**
University of Texas at Austin
rofuyu@cs.utexas.edu

**Nikhil Rao**
Technicolor Research
nikhilrao86@gmail.com

**Inderjit S. Dhillon**
University of Texas at Austin
inderjit@cs.utexas.edu

## Abstract

Time series prediction problems are becoming increasingly high-dimensional in modern applications, such as climatology and demand forecasting. For example, in the latter problem, the number of items for which demand needs to be forecast might be as large as 50,000. In addition, the data is generally noisy and full of missing values. Thus, modern applications require methods that are highly scalable, and can deal with noisy data in terms of corruptions or missing values. However, classical time series methods usually fall short of handling these issues. In this paper, we present a temporal regularized matrix factorization (TRMF) framework which supports data-driven temporal learning and forecasting. We develop novel regularization schemes and use scalable matrix factorization methods that are eminently suited for high-dimensional time series data that has many missing values. Our proposed TRMF is highly general, and subsumes many existing approaches for time series analysis. We make interesting connections to graph regularization methods in the context of learning the dependencies in an autoregressive framework. Experimental results show the superiority of TRMF in terms of scalability and prediction quality. In particular, TRMF is two orders of magnitude faster than other methods on a problem of dimension 50,000, and generates better forecasts on real-world datasets such as Wal-mart E-commerce datasets.

## 1 Introduction

Time series analysis is a central problem in many applications such as demand forecasting and climatology. Often, such applications require methods that are highly scalable to handle a very large number ($n$) of possibly inter-dependent one-dimensional time series and/or have a large time frame ($T$). For example, climatology applications involve data collected from possibly thousands of sensors, every hour (or less) over several years. Similarly, a store tracking its inventory would track thousands of items every day for multiple years. Not only is the scale of such problems huge, but they might also involve missing values, due to sensor malfunctions, occlusions or simple human errors. Thus, modern time series applications present two challenges to practitioners: scalability to handle large $n$ and $T$ and the flexibility to handle missing values.

Most approaches in the traditional time series literature such as autoregressive (AR) models or dynamic linear models (DLM)[7, 21] focus on low-dimensional time-series data and fall short of handling the two aforementioned issues. For example, an AR model of order $L$ requires $O(TL^2n^4 + L^3n^6)$ time to estimate $O(Ln^2)$ parameters, which is prohibitive even for moderate values of $n$. Similarly, Kalman filter based DLM approaches need $O(kn^2T + k^3T)$ computation cost to update parameters, where $k$ is the latent dimensionality, which is usually chosen to be larger than $n$ in many situations [13]. As a specific example, the maximum likelihood estimator implementation in the widely used R-DLM package [12], which relies on a general optimization solver, cannot scale beyond

$n$ in the tens. (See Appendix D for details). On the other hand, for models such as AR, the flexibility to handle missing values can also be very challenging even for one-dimensional time series [1], let alone the difficulty to handle high dimensional time series.

A natural way to model high-dimensional time series data is in the form of a matrix, with rows corresponding to each one-dimensional time series and columns corresponding to time points. In light of the observation that $n$ time series are usually highly correlated with each other, there have been some attempts to apply low-rank matrix factorization (MF) or matrix completion (MC) techniques to analyze high-dimensional time series [2, 14, 16, 23, 26]. Unlike the AR and DLM models above, state-of-the-art MF methods *scale linearly* in $n$, and hence can handle large datasets. Let $Y \in \mathbb{R}^{n \times T}$ be the matrix for the observed $n$-dimensional time series with $Y_{it}$ being the observation at the $t$-th time point of the $i$-th time series. Under the standard MF approach, $Y_{it}$ is estimated by the inner product $\boldsymbol{f}_i^\top \boldsymbol{x}_t$, where $\boldsymbol{f}_i \in \mathbb{R}^k$ is a $k$-dimensional latent embedding for the $i$-th time series, and $\boldsymbol{x}_t \in \mathbb{R}^k$ is a $k$-dimensional latent temporal embedding for the $t$-th time point. We can stack the $\boldsymbol{x}_t$s into the columns into a matrix $X \in \mathbb{R}^{k \times T}$ and $\boldsymbol{f}_i^\top$ into the rows of $F \in \mathbb{R}^{n \times k}$ (Figure 1) to get $Y \approx FX$. We can then solve:

$$\min_{F,X} \sum_{(i,t) \in \Omega} \left( Y_{it} - \boldsymbol{f}_i^\top \boldsymbol{x}_t \right)^2 + \lambda_f \mathcal{R}_f(F) + \lambda_x \mathcal{R}_x(X), \tag{1}$$

where $\Omega$ is the set of the observed entries. $\mathcal{R}_f(F)$, $\mathcal{R}_x(X)$ are regularizers for $F$ and $X$, which usually play a role to avoid overfitting and/or to encourage some specific *temporal structures* among the embeddings. It is clear that the common choice of the regularizer $\mathcal{R}_x(X) = \|X\|_F$ is no longer appropriate for time series applications, as it does not take into account the ordering among the temporal embeddings $\{\boldsymbol{x}_t\}$. Most existing MF approaches [2, 14, 16, 23, 26] adapt graph-based approaches to handle temporal dependencies. Specif-

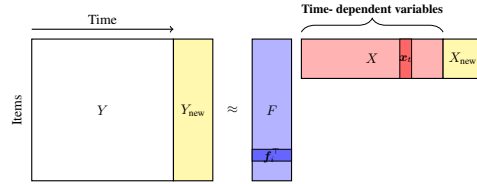

Figure 1: Matrix Factorization model for multiple time series. $F$ captures features for each time series in the matrix $Y$, and $X$ captures the latent and time-varying variables.

ically, the dependencies are described by a weighted similarity graph and incorporated through a Laplacian regularizer [18]. However, graph-based regularization fails in cases where there are negative correlations between two time points. Furthermore, unlike scenarios where explicit graph information is available with the data (such as a social network or product co-purchasing graph for recommender systems), explicit temporal dependency structure is usually unavailable and has to be inferred or approximated, which causes practitioners to either perform a separate procedure to estimate the dependencies or consider very short-term dependencies with simple fixed weights. Moreover, existing MF approaches, while yielding good estimations for missing values in past points, are poor in terms of forecasting future values, which is the problem of interest in time series analysis.

In this paper, we propose a novel temporal regularized matrix factorization framework (TRMF) for high-dimensional time series analysis. In TRMF, we consider a principled approach to **describe** the structure of temporal dependencies among latent temporal embeddings $\{\boldsymbol{x}_t\}$ and design a temporal regularizer to **incorporate** this temporal structure into the standard MF formulation. Unlike most existing MF approaches, our TRMF method supports data-driven temporal dependency learning and also brings the ability to **forecast** future values to a matrix factorization approach. In addition, inherited from the property of MF approaches, TRMF can easily handle high-dimensional time series data even in the presence of many missing values. As a specific example, we demonstrate a novel autoregressive temporal regularizer which encourages AR structure among temporal embeddings $\{\boldsymbol{x}_t\}$. We also make connections between the proposed regularization framework and graph-based approaches [18], where even negative correlations can be accounted for. This connection not only leads to better understanding about the dependency structure incorporated by our framework but also brings the benefit of using off-the-shelf efficient solvers such as GRALS [15] directly to solve TRMF.

**Paper Organization.** In Section 2, we review the existing approaches and their limitations on data with temporal dependencies. We present the proposed TRMF framework in Section 3, and show that the method is highly general and can be used for a variety of time series applications. We introduce a novel AR temporal regularizer in Section 4, and make connections to graph-based regularization approaches. We demonstrate the superiority of the proposed approach via extensive experimental results in Section 5 and conclude the paper in Section 6.

## 2 Motivations: Existing Approaches and Limitations

### 2.1 Classical Time-Series Models

Models such as AR and DLM are not suitable for modern multiple high-dimensional time series data (i.e., both $n$ and $T$ are large) due to their inherent computational inefficiency (see Section 1). To avoid overfitting in AR models, there have been studies with various structured transition matrices such as low rank and sparse matrices [5, 10, 11]. The focus of this research has been on obtaining better statistical guarantees. The scalability issue of AR models remains open. On the other hand, it is also challenging for many classic time-series models to deal with data that has many missing values [1].

In many situations where the model parameters are either given or designed by practitioners, the Kalman filter approach is used to perform forecasting, while the Kalman smoothing approach is used to impute missing entries. When model parameters are unknown, EM algorithms are applied to estimate both the model parameters and latent embeddings for DLM [3, 8, 9, 17, 19]. As most EM approaches for DLM contain the Kalman filter as a building block, they cannot scale to very high dimensional time series data. Indeed, as shown in Section 5, the popular R package for DLM's does not scale beyond data with tens of dimensions.

### 2.2 Existing Matrix Factorization Approaches for Data with Temporal Dependencies

In standard MF (1), the squared Frobenius norm $\mathcal{R}_x(X) = \|X\|_F^2 = \sum_{t=1}^T \|\boldsymbol{x}_t\|^2$ is usually the regularizer of choice for $X$. Because squared Frobenius norm assumes no dependencies among $\{\boldsymbol{x}_t\}$, standard MF formulation is *invariant to column permutation* and not applicable to data with temporal dependencies. Hence most existing temporal MF approaches turn to the framework of graph-based regularization [18] for temporally dependent $\{\boldsymbol{x}_t\}$, with a graph encoding the temporal dependencies. An exception is the work in [22], where the authors use specially designed regularizers to encourage a log-normal structure on the temporal coefficients.

**Graph regularization for temporal dependencies:** The framework of graph-based regularization is an approach to describe and incorporate general dependencies among variables. Let $G$ be a graph over $\{\boldsymbol{x}_t\}$ and $G_{ts}$ be the edge weight between the $t$-th node and $s$-th node. A popular regularizer to include as part of an objective function is the following:

$$\mathcal{R}_x(X) = \mathcal{G}(X \mid G, \eta) := \frac{1}{2} \sum_{t \sim s} G_{ts} \|\boldsymbol{x}_t - \boldsymbol{x}_s\|^2 + \frac{\eta}{2} \sum_t \|\boldsymbol{x}_t\|^2, \tag{2}$$

where $t \sim s$ denotes an edge between $t$-th node and $s$-th node, and the second summation term is used to guarantee strong convexity. A large $G_{ts}$ will ensure that $\boldsymbol{x}_t$ and $\boldsymbol{x}_s$ are close to each other in Euclidean distance, when (2) is minimized. Note that to guarantee the convexity of $\mathcal{G}(X \mid G, \eta)$, we need $G_{ts} \geq 0$.

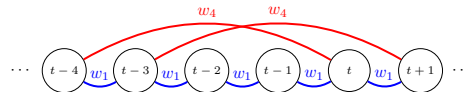

Figure 2: Graph-based regularization for temporal dependencies.

To apply graph-based regularizers to *temporal dependencies*, we need to specify the (repeating) dependency pattern by a lag set $\mathcal{L}$ and a weight vector $\boldsymbol{w}$ such that all the edges $t \sim s$ of distance $l$ (i.e., $|s - t| = l$) share the same weight $G_{ts} = w_l$. See Figure 2 for an example with $\mathcal{L} = \{1, 4\}$. Given $\mathcal{L}$ and $\boldsymbol{w}$, the corresponding graph regularizer becomes

$$\mathcal{G}(X \mid G, \eta) = \frac{1}{2} \sum_{l \in \mathcal{L}} \sum_{t:t>l} w_l (\boldsymbol{x}_t - \boldsymbol{x}_{t-l})^2 + \frac{\eta}{2} \sum_t \|\boldsymbol{x}_t\|^2. \tag{3}$$

This direct use of graph-based approach, while intuitive, has two issues: **a)** there might be negatively correlated dependencies between two time points; **b)** unlike many applications where such regularizers are used, the explicit temporal dependency structure is usually not available and has to be inferred. As a result, most existing approaches consider only very simple temporal dependencies such as a small size of $\mathcal{L}$ (e.g., $\mathcal{L} = \{1\}$) and/or uniform weights (e.g., $w_l = 1$, $\forall l \in \mathcal{L}$). For example, a simple chain graph is considered to design the smoothing regularizer in TCF [23]. This leads to poor forecasting abilities of existing MF methods for large-scale time series applications.

### 2.3 Challenges to Learn Temporal Dependencies

One could try to learn the weights $w_l$ automatically, by using the same regularizer as in (3) but with the weights unknown. This would lead to the following optimization problem:

$$\min_{F, X, \boldsymbol{w} \geq \boldsymbol{0}} \sum_{(i,t) \in \Omega} \left(Y_{it} - \boldsymbol{f}_i^\top \boldsymbol{x}_t\right)^2 + \lambda_f \mathcal{R}_f(F) + \frac{\lambda_x}{2} \sum_{l \in \mathcal{L}} \sum_{t:t-l>0} w_l (\boldsymbol{x}_t - \boldsymbol{x}_{t-l})^2 + \frac{\lambda_x \eta}{2} \sum_t \|\boldsymbol{x}_t\|^2, \tag{4}$$

where $\boldsymbol{0}$ is the zero vector, and $\boldsymbol{w} \geq \boldsymbol{0}$ is the constraint imposed by graph regularization.

It is not hard to see that the above optimization yields the trivial all-zero solution for $\boldsymbol{w}^*$, meaning the objective function is minimized when no temporal dependencies exist! To avoid the all zero solution, one might want to impose a simplex constraint on $\boldsymbol{w}$ (i.e., $\sum_{l\in\mathcal{L}} w_l = 1$). Again, it is not hard to see that this will result in $\boldsymbol{w}^*$ being a 1-sparse vector, with $w_{l^*}$ being 1, where $l^* = \arg\min_{l\in\mathcal{L}} \sum_{t:t>l} \|\boldsymbol{x}_t - \boldsymbol{x}_{t-l}\|^2$. Thus, looking to learn the weights automatically by simply plugging in the regularizer in the MF formulation is not a viable option.

# 3   Temporal Regularized Matrix Factorization

In order to resolve the limitations mentioned in Sections 2.2 and 2.3, we propose the **T**emporal **R**egularized **M**atrix **F**actorization (TRMF) framework, which is a novel approach to incorporate temporal dependencies into matrix factorization models. Unlike the aforementioned graph-based approaches, we propose to use well-studied time series models to describe temporal dependencies among $\{\boldsymbol{x}_t\}$ explicitly. Such models take the form:

$$\boldsymbol{x}_t = M_\Theta(\{\boldsymbol{x}_{t-l} : l \in \mathcal{L}\}) + \boldsymbol{\epsilon}_t, \tag{5}$$

where $\boldsymbol{\epsilon}_t$ is a Gaussian noise vector, and $M_\Theta$ is the time-series model parameterized by $\mathcal{L}$ and $\Theta$. $\mathcal{L}$ is a set containing the lag indices $l$, denoting a dependency between $t$-th and $(t-l)$-th time points, while $\Theta$ captures the weighting information of temporal dependencies (such as the transition matrix in AR models). To incorporate the temporal dependency into the standard MF formulation (1), we propose to design a new regularizer $\mathcal{T}_\text{M}(X \mid \Theta)$ which encourages the structure induced by $M_\Theta$.

Taking a standard approach to model time series, we set $\mathcal{T}_\text{M}(X \mid \Theta)$ be the negative log likelihood of observing a particular realization of the $\{\boldsymbol{x}_t\}$ for a given model $M_\Theta$:

$$\mathcal{T}_\text{M}(X \mid \Theta) = -\log \mathbb{P}(\boldsymbol{x}_1, \ldots, \boldsymbol{x}_T \mid \Theta). \tag{6}$$

When $\Theta$ is given, we can use $\mathcal{R}_x(X) = \mathcal{T}_\text{M}(X \mid \Theta)$ in the MF formulation (1) to encourage $\{\boldsymbol{x}_t\}$ to follow the temporal dependency induced by $M_\Theta$. When the $\Theta$ is unknown, we can treat $\Theta$ as another set of variables and include another regularizer $\mathcal{R}_\theta(\Theta)$ into (1):

$$\min_{F,X,\Theta} \quad \sum_{(i,t)\in\Omega} \left(Y_{it} - \boldsymbol{f}_i^\top \boldsymbol{x}_t\right)^2 + \lambda_f \mathcal{R}_f(F) + \lambda_x \mathcal{T}_\text{M}(X \mid \Theta) + \lambda_\theta \mathcal{R}_\theta(\Theta), \tag{7}$$

which be solved by an alternating minimization procedure over $F$, $X$, and $\Theta$.

**Data-driven Temporal Dependency Learning in TRMF:** Recall that in Section 2.3, we showed that directly using graph based regularizers to incorporate temporal dependencies leads to trivial solutions for the weights. TRMF circumvents this issue. When $F$ and $X$ are fixed, (7) is reduced to:

$$\min_\Theta \quad \lambda_x \mathcal{T}_\text{M}(X \mid \Theta) + \lambda_\theta \mathcal{R}_\theta(\Theta), \tag{8}$$

which is a maximum-a-posterior (MAP) estimation problem (in the Bayesian sense) to estimate the best $\Theta$ for a given $\{\boldsymbol{x}_t\}$ under the $M_\Theta$ model. There are well-developed algorithms to solve (8) and obtain non-trivial $\Theta$. Thus, unlike most existing temporal matrix factorization approaches where the strength of dependencies is fixed, $\Theta$ in TRMF can be learned automatically from data.

**Time Series Analysis with TRMF:** We can see that TRMF (7) lends itself seamlessly to handle a variety of commonly encountered tasks in analyzing data with temporal dependency:
- **Time-series Forecasting:** Once we have $M_\Theta$ for latent embeddings $\{\boldsymbol{x}_t : 1, \ldots, T\}$, we can use it to predict future latent embeddings $\{\boldsymbol{x}_t : t > T\}$ and have the ability to obtain non-trivial forecasting results for $\boldsymbol{y}_t = F\boldsymbol{x}_t$ for $t > T$.
- **Missing-value Imputation:** In some time-series applications, some entries in $Y$ might be unobserved, for example, due to faulty sensors in electricity usage monitoring or occlusions in the case of motion recognition in video. We can use $\boldsymbol{f}_i^\top \boldsymbol{x}_t$ to impute these missing entries, much like standard matrix completion, and is useful in recommender systems [23] and sensor networks [26].

**Extensions to Incorporate Extra Information:** Like matrix factorization, TRMF (7) can be extended to incorporate additional information. For example, pairwise relationships between the time series can be incorporated using structural regularizers on $F$. Furthermore, when features are known for the time series, we can make use of interaction models such as those in [6, 24, 25]. Also, TRMF can be extended to tensors. More details on these extensions can be found in Appendix B.

# 4   A Novel Autoregressive Temporal Regularizer

In Section 3, we described the TRMF framework in a very general sense, with the regularizer $\mathcal{T}_\text{M}(X \mid \Theta)$ incorporating dependencies specified by the time series model $M_\Theta$. In this section, we specialize this to the case of AR models, which are parameterized by a lag set $\mathcal{L}$ and weights $\mathcal{W} = \left\{W^{(l)} \in \mathbb{R}^{k\times k} : l \in \mathcal{L}\right\}$. Assume that $\boldsymbol{x}_t$ is a noisy linear combination of some previous

points; that is, $\boldsymbol{x}_t = \sum_{l \in \mathcal{L}} W^{(l)} \boldsymbol{x}_{t-l} + \boldsymbol{\epsilon}_t$, where $\boldsymbol{\epsilon}_t$ is a Gaussian noise vector. For simplicity, we assume that the $\boldsymbol{\epsilon}_t \sim \mathcal{N}(0, \sigma^2 I_k)$, where $I_k$ is the $k \times k$ identity matrix[1]. The temporal regularizer $\mathcal{T}_M(X \mid \Theta)$ corresponding to this AR model can be written as:

$$\mathcal{T}_{AR}(X \mid \mathcal{L}, \mathcal{W}, \eta) := \frac{1}{2} \sum_{t=m}^{T} \left\| \boldsymbol{x}_t - \sum_{l \in \mathcal{L}} W^{(l)} \boldsymbol{x}_{t-l} \right\|^2 + \frac{\eta}{2} \sum_t \|\boldsymbol{x}_t\|^2, \tag{9}$$

where $m := 1 + L$, $L := \max(\mathcal{L})$, and $\eta > 0$ to guarantee the strong convexity of (9).

TRMF allows us to learn the weights $\{W^{(l)}\}$ when they are unknown. Since each $W^{(l)} \in \mathbb{R}^{k \times k}$, there will be $|\mathcal{L}|k^2$ variables to learn, which may lead to overfitting. To prevent this and to yield more interpretable results, we consider diagonal $W^{(l)}$, reducing the number of parameters to $|\mathcal{L}|k$. To simplify notation, we use $\mathcal{W}$ to denote the $k \times L$ matrix where the $l$-th column constitutes the diagonal elements of $W^{(l)}$. Note that for $l \notin \mathcal{L}$, the $l$-th column of $\mathcal{W}$ is a zero vector. Let $\bar{\boldsymbol{x}}_r^\top = [\cdots, X_{rt}, \cdots]$ be the $r$-th row of $X$ and $\bar{\boldsymbol{w}}_r^\top = [\cdots, \mathcal{W}_{rl}, \cdots]$ be the $r$-th row of $\mathcal{W}$. Then (9) can be written as $\mathcal{T}_{AR}(X \mid \mathcal{L}, \mathcal{W}, \eta) = \sum_{r=1}^k \mathcal{T}_{AR}(\bar{\boldsymbol{x}}_r \mid \mathcal{L}, \bar{\boldsymbol{w}}_r, \eta)$, where we define

$$\mathcal{T}_{AR}(\bar{\boldsymbol{x}} \mid \mathcal{L}, \bar{\boldsymbol{w}}, \eta) = \frac{1}{2} \sum_{t=m}^{T} \left( x_t - \sum_{l \in \mathcal{L}} w_l x_{t-l} \right)^2 + \frac{\eta}{2} \|\bar{\boldsymbol{x}}\|^2, \tag{10}$$

with $x_t$ being the $t$-th element of $\bar{\boldsymbol{x}}$, and $w_l$ being the $l$-th element of $\bar{\boldsymbol{w}}$.

**Correlations among Multiple Time Series.** Even when $\{W^l\}$ is diagonal, TRMF retains the power to capture the correlations among time series via the factors $\{\boldsymbol{f}_i\}$, since it has an effect only on the structure of latent embeddings $\{\boldsymbol{x}_t\}$. Indeed, as the $i$-th dimension of $\{\boldsymbol{y}_t\}$ is modeled by $\boldsymbol{f}_i^\top X$ in (7), the low rank $F$ is a $k$ dimensional latent embedding of multiple time series. This embedding captures correlations among multiple time series. Furthermore, $\{\boldsymbol{f}_i\}$ acts as time series features, which can be used to perform classification/clustering even in the presence of missing values.

**Choice of Lag Index Set $\mathcal{L}$.** Unlike most approaches mentioned in Section 2.2, the choice of $\mathcal{L}$ in TRMF is more flexible. Thus, TRMF can provide important advantages: First, because there is no need to specify the weight parameters $\mathcal{W}$, $\mathcal{L}$ can be chosen to be larger to account for long range dependencies, which also yields more accurate and robust forecasts. Second, the indices in $\mathcal{L}$ can be discontinuous so that one can easily embed *domain knowledge* about periodicity or seasonality. For example, one might consider $\mathcal{L} = \{1, 2, 3, 51, 52, 53\}$ for weekly data with a one year seasonality.

**Connections to Graph Regularization.** We now establish connections between $\mathcal{T}_{AR}(\bar{\boldsymbol{x}} \mid \mathcal{L}, \bar{\boldsymbol{w}}, \eta)$ and graph regularization (2) for matrix factorization. Let $\bar{\mathcal{L}} := \mathcal{L} \cup \{0\}$, $w_0 = -1$ so that (10) is

$$\mathcal{T}_{AR}(\bar{\boldsymbol{x}} \mid \mathcal{L}, \bar{\boldsymbol{w}}, \eta) = \frac{1}{2} \sum_{t=m}^{T} \left( \sum_{l \in \bar{\mathcal{L}}} w_l x_{t-l} \right)^2 + \frac{\eta}{2} \|\bar{\boldsymbol{x}}\|^2,$$

and let $\delta(d) := \{l \in \bar{\mathcal{L}} : l - d \in \bar{\mathcal{L}}\}$. We then have the following result:

**Theorem 1.** *Given a lag index set $\mathcal{L}$, weight vector $\bar{\boldsymbol{w}} \in \mathbb{R}^L$, and $\bar{\boldsymbol{x}} \in \mathbb{R}^T$, there is a weighted signed graph $G^{AR}$ with $T$ nodes and a diagonal matrix $D \in \mathbb{R}^{T \times T}$ such that*

$$\mathcal{T}_{AR}(\bar{\boldsymbol{x}} \mid \mathcal{L}, \bar{\boldsymbol{w}}, \eta) = \mathcal{G}\big(\bar{\boldsymbol{x}} \mid G^{AR}, \eta\big) + \frac{1}{2} \bar{\boldsymbol{x}}^\top D \bar{\boldsymbol{x}}, \tag{11}$$

*where $\mathcal{G}\big(\bar{\boldsymbol{x}} \mid G^{AR}, \eta\big)$ is the graph regularization (2) with $G = G^{AR}$. Furthermore, $\forall t$ and $d$*

$$G_{t,t+d}^{AR} = \begin{cases} \displaystyle\sum_{l \in \delta(d)} \sum_{m \le t+l \le T} -w_l w_{l-d} & \text{if } \delta(d) \ne \phi, \\ 0 & \text{otherwise,} \end{cases} \quad \text{and } D_{tt} = \left( \sum_{l \in \bar{\mathcal{L}}} w_l \right) \left( \sum_{l \in \bar{\mathcal{L}}} w_l [m \le t+l \le T] \right)$$

See Appendix C.1 for a detailed proof. From Theorem 1, we see that $\delta(d)$ is non-empty if and only if there are edges between time points separated by $d$ in $G^{AR}$. Thus, we can construct the dependency graph for $\mathcal{T}_{AR}(\bar{\boldsymbol{x}} \mid \mathcal{L}, \bar{\boldsymbol{w}}, \eta)$ by checking whether $\delta(d)$ is empty. Figure 3 demonstrates an example with $\mathcal{L} = \{1, 4\}$. We can see that besides edges of distance $d = 1$ and $d = 4$, there are also edges of distance $d = 3$ (dotted edges in Figure 3) because $4 - 3 \in \bar{\mathcal{L}}$ and $\delta(3) = \{4\}$.

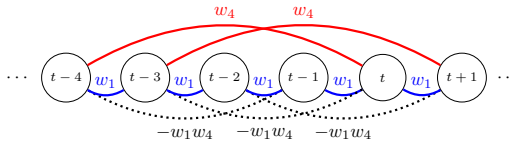

Figure 3: The graph structure induced by the AR temporal regularizer (10) with $\mathcal{L} = \{1, 4\}$.

Table 1: Data statistics.

|  | synthetic | electricity | traffic | walmart-1 | walmart-2 |
|---|---|---|---|---|---|
| $n$ | 16 | 370 | 963 | 1,350 | 1,582 |
| $T$ | 128 | 26,304 | 10,560 | 187 | 187 |
| missing ratio | 0% | 0% | 0% | 55.3% | 49.3% |

Although Theorem 1 shows that AR-based regularizers are similar to the graph-based regularization framework, we note the following key differences:

- The graph $G^{\text{AR}}$ in Theorem 1 contains **both** positive and negative edges. This implies that the AR temporal regularizer is able to support negative correlations, which the standard graph-based regularizer cannot. This can make $\mathcal{G}\left(\bar{\boldsymbol{x}} \mid G^{\text{AR}}, \eta\right)$ non-convex. The addition of the second term in (11), however, still leads to a convex regularizer $\mathcal{T}_{\text{AR}}(\bar{\boldsymbol{x}} | \mathcal{L}, \bar{\boldsymbol{w}}, \eta)$.
- Unlike (3) where there is freedom to specify a weight for each distance, in the graph $G^{\text{AR}}$, the weight values for the edges are **more structured** (e.g., the weight for $d = 3$ in Figure 3 is $-w_1 w_4$). Hence, minimization w.r.t. $w's$ is not trivial, and neither are the obtained solutions.

Plugging $\mathcal{T}_{\text{M}}(X \mid \Theta) = \mathcal{T}_{\text{AR}}(X | \mathcal{L}, \mathcal{W}, \eta)$ into (7), we obtain the following problem:

$$\min_{F,X,\mathcal{W}} \sum_{(i,t)\in\Omega} \left(Y_{it} - \boldsymbol{f}_i^\top \boldsymbol{x}_t\right)^2 + \lambda_f \mathcal{R}_f(F) + \sum_{r=1}^{k} \lambda_x \mathcal{T}_{\text{AR}}(\bar{\boldsymbol{x}}_r | \mathcal{L}, \bar{\boldsymbol{w}}_r, \eta) + \lambda_w \mathcal{R}_w(\mathcal{W}), \quad (12)$$

where $\mathcal{R}_w(\mathcal{W})$ is a regularizer for $\mathcal{W}$. We will refer to (12) as TRMF-AR. We can apply alternating minimization to solve (12). In fact, solving for each variable reduces to well known methods, for which highly efficient algorithms exist:

**Updates for $F$.** When $X$ and $\mathcal{W}$ are fixed, the subproblem of updating $F$ is the same as updating $F$ while $X$ fixed in (1). Thus, fast algorithms such as alternating least squares or coordinate descent can be applied directly to find $F$, which costs $O(|\Omega|k^2)$ time.

**Updates for $X$.** We solve $\arg\min_X \sum_{(i,t)\in\Omega}\left(Y_{it} - \boldsymbol{f}_i^\top \boldsymbol{x}_t\right)^2 + \lambda_x \sum_{r=1}^{k} \mathcal{T}_{\text{AR}}(\bar{\boldsymbol{x}}_r | \mathcal{L}, \bar{\boldsymbol{w}}_r, \eta)$. From Theorem 1, we see that $\mathcal{T}_{\text{AR}}(\bar{\boldsymbol{x}} | \mathcal{L}, \bar{\boldsymbol{w}}, \eta)$ shares the same form as the graph regularizer, and we can apply GRALS [15] to find $X$, which costs $O(|\mathcal{L}|Tk^2)$ time.

**Updates for $\mathcal{W}$.** How to update $\mathcal{W}$ while $F$ and $X$ fixed depends on the choice of $\mathcal{R}_w(\mathcal{W})$. There are many parameter estimation techniques developed for AR with various regularizers [11, 20]. For simplicity, we consider the squared Frobenius norm: $\mathcal{R}_w(\mathcal{W}) = \|\mathcal{W}\|_F^2$. As a result, each row of $\bar{\boldsymbol{w}}_r$ of $\mathcal{W}$ can be updated by solving the following one-dimensional autoregressive problem.

$$\arg\min_{\bar{\boldsymbol{w}}} \; \lambda_x \mathcal{T}_{\text{AR}}(\bar{\boldsymbol{x}}_r | \mathcal{L}, \bar{\boldsymbol{w}}, \eta) + \lambda_w \|\bar{\boldsymbol{w}}\|^2 \equiv \arg\min_{\bar{\boldsymbol{w}}} \; \sum_{t=m}^{T}\left(x_t - \sum_{l\in\mathcal{L}} w_l x_{t-l}\right)^2 + \frac{\lambda_w}{\lambda_x}\|\bar{\boldsymbol{w}}\|^2,$$

which is a simple $|\mathcal{L}|$ dimensional ridge regression problem with $T - m + 1$ instances, which can be solved efficiently by Cholesky factorization in $O(|\mathcal{L}|^3 + T|\mathcal{L}|^2)$ time

Note that since our method is highly modular, one can resort to *any* method to solve the optimization subproblems that arise for each module. Moreover, as mentioned in Section 3, TRMF can also be used with different regularization structures, making it highly adaptable.

### 4.1 Connections to Existing MF Approaches
TRMF-AR is a generalization of many existing MF approaches to handle data with temporal dependencies. Specifically, Temporal Collaborative Filtering [23] corresponds to $W^{(1)} = I_k$ on $\{\boldsymbol{x}_t\}$. The NMF method of [2] is an AR($L$) model with $W^{(l)} = \alpha^{l-1}(1 - \alpha)I_k$, $\forall l$, where $\alpha$ is pre-defined. The AR(1) model of [16, 26] has $W^{(1)} = I_n$ on $\{F\boldsymbol{x}_t\}$. Finally the DLM [7] is a latent AR(1) model with a general $W^{(1)}$, which can be estimated by EM algorithms.

### 4.2 Connections to Learning Gaussian Markov Random Fields
The Gaussian Markov Random Field (GMRF) is a general way to model multivariate data with dependencies. GMRF assumes that data are generated from a multivariate Gaussian distribution with a covariance matrix $\Sigma$ which describes the dependencies among $T$ dimensional variables i.e., $\bar{\boldsymbol{x}} \sim \mathcal{N}(0, \Sigma)$. If the unknown $\bar{\boldsymbol{x}}$ is assumed to be generated from this model, The negative log likelihood of the data can be written as $\bar{\boldsymbol{x}}^\top \Sigma^{-1} \bar{\boldsymbol{x}}$, ignoring the constants and where $\Sigma^{-1}$ is the inverse covariance matrix of the Gaussian distribution. This prior can be incorporated into an empirical risk minimization framework as a regularizer. Furthermore, it is known that if $\left(\Sigma^{-1}\right)_{st} = 0$, $x_t$ and $x_s$ are conditionally independent, given the other variables. In Theorem 1 we established connections

Table 2: Forecasting results: ND/ NRMSE for each approach. Lower values are better. "-" indicates an unavailability due to scalability or an inability to handle missing values.

| | Forecasting with Full Observation | | | | | | |
| | Matrix Factorization Models | | | Time Series Models | | | |
| | TRMF-AR | SVD-AR(1) | TCF | AR(1) | DLM | R-DLM | Mean |
|---|---|---|---|---|---|---|---|
| synthetic | **0.373/ 0.487** | 0.444/ 0.872 | 1.000/ 1.424 | 0.928/ 1.401 | 0.936/ 1.391 | 0.996/ 1.420 | 1.000/ 1.424 |
| electricity | 0.255/ **1.397** | 0.257/ 1.865 | 0.349/ 1.838 | **0.219/** 1.439 | 0.435/ 2.753 | -/ - | 1.410/ 4.528 |
| traffic | **0.187/ 0.423** | 0.555/ 1.194 | 0.624/ 0.931 | 0.275/ 0.536 | 0.639/ 0.951 | -/ - | 0.560/ 0.826 |
| | Forecasting with Missing Values | | | | | | |
| walmart-1 | **0.533/ 1.958** | -/ - | 0.540/2.231 | -/ - | 0.602/ 2.293 | -/ - | 1.239/3.103 |
| walmart-2 | **0.432/ 1.065** | -/ - | 0.446/1.124 | -/ - | 0.453/ 1.110 | -/ - | 1.097/2.088 |

to graph based regularizers, and that such methods can be seen as regularizing with the inverse covariance matrix for Gaussians [27]. We thus have the following result:

**Corollary 1.** *For any lag set $\mathcal{L}$, $\bar{w}$, and $\eta > 0$, the inverse covariance matrix $\Sigma_{AR}^{-1}$ of the GMRF model corresponding to the quadratic regularizer $\mathcal{R}_x(\bar{x}) := \mathcal{T}_{AR}(\bar{x}|\mathcal{L}, \bar{w}, \eta)$ shares the same off-diagonal non-zero pattern as $G^{AR}$ defined in Theorem 1. Moreover, we have $\mathcal{T}_{AR}(\bar{x}|\mathcal{L}, \bar{w}, \eta) = \bar{x}^\top \Sigma_{AR}^{-1} \bar{x}$.*

A detailed proof is in Appendix C.2. As a result, our proposed AR-based regularizer is equivalent to imposing a Gaussian prior on $\bar{x}$ with a structured inverse covariance described by the matrix $G^{AR}$ defined in Theorem 1. Moreover, the step to learn $\mathcal{W}$ has a natural interpretation: the lag set $\mathcal{L}$ imposes the non-zero pattern of the graphical model on the data, and then we solve a simple least squares problem to learn the weights corresponding to the edges. As an application of Theorem 1 from [15] and Corollary 1, when $\mathcal{R}_f(F) = \|F\|_F^2$, we can relate $\mathcal{T}_{AR}$ to a weighted nuclear norm:

$$\|ZB\|_* = \frac{1}{2} \inf_{F, X:Z=FX} \|F\|_F^2 + \sum_r \mathcal{T}_{AR}(\bar{x}_r|\mathcal{L}, \bar{w}, \eta), \tag{13}$$

where $B = US^{1/2}$ and $\Sigma_{AR}^{-1} = USU^\top$ is the eigen-decomposition of $\Sigma_{AR}^{-1}$. (13) enables us to apply the results from [15] to obtain guarantees for the use of AR temporal regularizer when $\mathcal{W}$ is given. For simplicity, we assume $\bar{w}_r = \bar{w}$, $\forall r$ and consider a relaxed convex formulation for (12) as follows:

$$\hat{Z} = \arg\min_{Z \in \mathcal{C}} \frac{1}{N} \sum_{(i,j)\in\Omega} (Y_{ij} - Z_{ij})^2 + \lambda_z \|ZB\|_*, \tag{14}$$

where $N = |\Omega|$, and $\mathcal{C}$ is a set of matrices with low *spikiness*. Full details are provided in Appendix C.3. As an application of Theorem 2 from [15], we have the following corollary.

**Corollary 2.** *Let $Z^\star = FX$ be the ground truth $n \times T$ time series matrix of rank $k$. Let $Y$ be the matrix with $N = |\Omega|$ randomly observed entries corrupted with additive Gaussian noise with variance $\sigma^2$. Then if $\lambda_z \geq C_1 \sqrt{\frac{(n+T)\log(n+T)}{N}}$, with high probability for the $\hat{Z}$ obtained by (14),*

$$\left\| Z^\star - \hat{Z} \right\|_F \leq C_2 \alpha^2 \max(1, \sigma^2) \frac{k(n+T)\log(n+T)}{N} + O(\alpha^2/N),$$

*where $C_1, C_2$ are positive constants, and $\alpha$ depends on the product $Z^\star B$.*

See Appendix C.3 for details. From the results in Table 3, we observe superior performance of TRMF-AR over standard MF, indicating that $\bar{w}$ learnt from our data-driven approach (12) does aid in recovering the missing entries for time series. We would like to point out that establishing a theoretical guarantee for TRMF with $\mathcal{W}$ is unknown remains a challenging research direction.

## 5 Experimental Results

We consider five datasets (Table 1). For synthetic, we first randomly generate $F \in \mathbb{R}^{16 \times 4}$ and generate $\{x_t\}$ following an AR process with $\mathcal{L} = \{1, 8\}$. Then $Y$ is obtained by $y_t = Fx_t + \epsilon_t$ where $\epsilon_t \sim \mathcal{N}(0, 0.1I)$. The data sets electricity and traffic are obtained from the UCI repository, while walmart-1 and walmart-2 are two propriety datasets from Walmart E-commerce containing weekly sale information. Due to reasons such as out-of-stock, 55.3% and 49.3% of entries are missing respectively. To evaluate the prediction performance, we consider the normalized deviation (ND) and normalized RMSE (NRMSE). See details for the description for each dataset and the formal definition for each criterion in Appendix A.

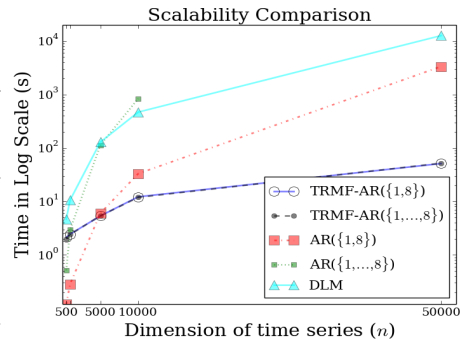

Figure 4: Scalability: $T = 512$. $n \in \{500, 1000, \ldots, 50000\}$. AR($\{1, \ldots, 8\}$) cannot finish in 1 day.

Table 3: Missing value imputation results: ND/ NRMSE for each approach. Note that TRMF outperforms all competing methods in almost all cases.

| | $\frac{|\Omega|}{n \times T}$ | Matrix Factorization Models | | | Time Series Models | |
|---|---|---|---|---|---|---|
| | | TRMF-AR | TCF | MF | DLM | Mean |
| synthetic | 20% | **0.467/ 0.661** | 0.713/ 1.030 | 0.688/ 1.064 | 0.933/ 1.382 | 1.002/ 1.474 |
| | 30% | **0.336/ 0.455** | 0.629/ 0.961 | 0.595/ 0.926 | 0.913/ 1.324 | 1.004/ 1.445 |
| | 40% | **0.231/ 0.306** | 0.495/ 0.771 | 0.374/ 0.548 | 0.834/ 1.259 | 1.002/ 1.479 |
| | 50% | **0.201/ 0.270** | 0.289/ 0.464 | 0.317/ 0.477 | 0.772/ 1.186 | 1.001/ 1.498 |
| electricity | 20% | **0.245/ 2.395** | 0.255/ 2.427 | 0.362/ 2.903 | 0.462/ 4.777 | 1.333/ 6.031 |
| | 30% | **0.235/ 2.415** | 0.245/ 2.436 | 0.355/ 2.766 | 0.410/ 6.605 | 1.320/ 6.050 |
| | 40% | 0.231/ 2.429 | 0.242/ 2.457 | 0.348/ 2.697 | **0.196/ 2.151** | 1.322/ 6.030 |
| | 50% | 0.223/ 2.434 | 0.233/ 2.459 | 0.319/ 2.623 | **0.158/ 1.590** | 1.320/ 6.109 |
| traffic | 20% | **0.190/ 0.427** | 0.208/ 0.448 | 0.310/ 0.604 | 0.353/ 0.603 | 0.578/ 0.857 |
| | 30% | **0.186/ 0.419** | 0.199/ 0.432 | 0.299/ 0.581 | 0.286/ 0.518 | 0.578/ 0.856 |
| | 40% | **0.185/ 0.416** | 0.198/ 0.428 | 0.292/ 0.568 | 0.251/ 0.476 | 0.578/ 0.857 |
| | 50% | **0.184/ 0.415** | 0.193/ 0.422 | 0.251/ 0.510 | 0.224/ 0.447 | 0.578/ 0.857 |

**Methods/Implementations Compared:**
- TRMF-AR: The proposed formulation (12) with $\mathcal{R}_w(\mathcal{W}) = \|\mathcal{W}\|_F^2$. For $\mathcal{L}$, we use $\{1, 2, \ldots, 8\}$ for synthetic, $\{1, \ldots, 24\} \cup \{7 \times 24, \ldots, 8 \times 24 - 1\}$ for electricity and traffic, and $\{1, \ldots, 10\} \cup \{50, \ldots, 56\}$ for walmart-1 and walmart-2 to capture seasonality.
- SVD-AR(1): The rank-$k$ approximation of $Y = USV^\top$ is first obtained by SVD. After setting $F = US$ and $X = V^\top$, a $k$-dimensional AR(1) is learned on $X$ for forecasting.
- TCF: Matrix factorization with the simple temporal regularizer proposed in [23].
- AR(1): $n$-dimensional AR(1) model.[2]
- DLM: two implementations: the widely used R-DLM package [12] and the code provided in [8].
- Mean: The baseline, which predicts everything to be the mean of the observed portion of $Y$.

For each method and data set, we perform a grid search over various parameters (such as $k$, $\lambda$ values) following a rolling validation approach described in [11].

**Scalability:** Figure 4 shows that traditional time-series approaches such as AR or DLM suffer from the scalability issue for large $n$, while TRMF-AR scales much better with n. Specifically, for $n = 50,000$, TRMF is 2 orders of magnitude faster than competing AR/DLM methods. Note that the results for R-DLM are not available because the R package cannot scale beyond $n$ in the tens (See Appendix D for more details.). Furthermore, the `dlmMLE` routine in R-DLM uses a general optimization solver, which is orders of magnitude slower than the implementation provided in [8].

## 5.1 Forecasting

**Forecasting with Full Observations.** We first compare various methods on the task of forecasting values in the test set, given fully observed training data. For synthetic, we consider one-point ahead forecasting task and use the last ten time points as the test periods. For electricity and traffic, we consider the 24-hour ahead forecasting task and use last seven days as the test periods. From Table 2, we can see that TRMF-AR outperforms all the other methods on both metrics considered.

**Forecasting with Missing Values.** We next compare the methods on the task of forecasting in the presence of missing values in the data. We use the Walmart datasets here, and consider 6-week ahead forecasting and use last 54 weeks as the test periods. Note that SVD-AR(1) and AR(1) cannot handle missing values. The second part of Table 2 shows that we again outperform other methods.

## 5.2 Missing Value Imputation

We next consider the case of imputing missing values in the data. As in [9], we assume that blocks of data are missing, corresponding to sensor malfunctions for example, over a length of time. To create data with missing entries, we first fixed the percentage of data that we were interested in observing, and then uniformly at random occluded blocks of a predetermined length (2 for synthetic data and 5 for the real datasets). The goal was to predict the occluded values. Table 3 shows that TRMF outperforms the methods we compared to on almost all cases.

## 6 Conclusions

We propose a novel temporal regularized matrix factorization framework (TRMF) for high-dimensional time series problems with missing values. TRMF not only models temporal dependency among the data points, but also supports data-driven dependency learning. TRMF generalizes several well-known methods and yields superior performance when compared to other state-of-the-art methods on real-world datasets.

**Acknowledgements:** This research was supported by NSF grants (CCF-1320746, IIS-1546459, and CCF-1564000) and gifts from Walmart Labs and Adobe. We thank Abhay Jha for the help on Walmart experiments.

## Footnotes

[1] If the (known) covariance matrix is not identity, we can suitably modify the regularizer.

[2]In Appendix A, we also show a baseline which applies an independent AR model to each dimension.

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
