[Supplementary Material]

# A  More Details about Experiments

## A.1  Datasets and Evaluation Criteria
**Datasets:**

- synthetic: a small synthetic dataset with $n = 16, T = 128$. We generate $\{x_t \in \mathbb{R}^4 : t = 1, \dots, 128\}$ from the autoregressive process with a lag index set $\mathcal{L} = \{1, 8\}$, randomly generated $\{W^{(l)}\}$, and an additive white Gaussian noise of $\sigma = 0.1$. We then randomly generate a matrix $F \in \mathbb{R}^{16 \times 4}$ and obtain $y_t = Fx_t + \epsilon_t$, where $\epsilon \sim \mathcal{N}(0, 0.1)$.
- electricity[3]: the electricity usage in kW recorded every 15 minutes, for $n = 370$ clients. We convert the data to reflect hourly consumption, by aggregating blocks of 4 columns, to obtain $T = 26,304$. Teh coefficient of variation for electricity is $6.0341$.
- traffic[4]: A collection of 15 months of daily data from the California Department of Transportation. The data describes the occupancy rate, between 0 and 1, of different car lanes of San Francisco bay area freeways. The data was sampled every 10 minutes, and we again aggregate the columns to obtain hourly traffic data to finally get $n = 963, T = 10,560$. The coefficient of variation for traffic is $0.8565$.
- walmart-1 & walmart-2: two propriety datasets from Walmart E-commerce contain weekly sale information of 1,350 and 1,582 items for 187 weeks, respectively. The time-series of sales for each item start and end at different time points; for modeling purposes we assume one start and end timestamp by padding each series with missing values. This along with some other missing values due to out-of-stock reasons lead to 55.3% and 49.3% of entries being missing.

**Evaluation Criteria:** We compute the normalized deviation (ND) and normalized RMSE (NRMSE).

$$\text{Normalized deviation (ND):} \quad \left( \frac{1}{|\Omega_{test}|} \sum_{(i,t) \in \Omega_{test}} \left| \hat{Y}_{it} - Y_{it} \right| \right) \Big/ \left( \frac{1}{|\Omega_{test}|} \sum_{(i,t) \in \Omega_{test}} |Y_{ij}| \right)$$

$$\text{Normalized RMSE (NRMSE):} \quad \sqrt{ \frac{1}{|\Omega_{test}|} \sum_{(i,t) \in \Omega_{test}} \left( \hat{Y}_{it} - Y_{it} \right)^2 } \Big/ \left( \frac{1}{|\Omega_{test}|} \sum_{(i,t) \in \Omega_{test}} |Y_{ij}| \right)$$

For each method and data set, we perform the grid search over various parameters (such as $k$, $\lambda$ values) following a rolling validation approach described in [11]. We search $k \in \{2, 4, 8\}$ for synthetic and $\in \{20, 40\}$ for other datasets. For TRMF-AR, SVD-AR(1), TCF, and AR(1), we search $\lambda \in \{50, 5, 0.5, 0.05\}$

**Results with $n$ independent AR models.** One naive way for AR models to solve the scalability issue to handle $n$-dimension time series is assuming each dimension is independent from each other. Thus, one can apply an one-dimensional AR model to each dimension. Although this approach is embarrassingly parallel, it ignores the correlations among time series that we can take into account. Note that this approach still cannot solve the difficulty of AR models to handle missing values. Here, we include the forecasting results of this independent AR approach in Table 4. For synthetic, we learn an independent AR(1) for each dimension, while for electricity and traffic, we learn an independent AR($\{1, 24\}$) for each dimension. We can see that on synthetic and traffic, TRMF-AR still outperforms this independent AR approach.

Table 4: Forecasting Comparison.

|  | TRMF-AR | AR(1) | Independent AR | Mean |
|---|---|---|---|---|
| synthetic | **0.373/ 0.487** | 0.928/ 1.401 | 0.973/ 1.388 | 1.000/ 1.424 |
| electricity | 0.255/ 1.397 | 0.219/ 1.439 | **0.206/1.220** | 1.410/ 4.528 |
| traffic | **0.187/ 0.423** | 0.275/ 0.536 | 0.294/ 0.591 | 0.560/ 0.826 |

# B  Extensions to Incorporate Extra Information

In the same vein as matrix factorization approaches, TRMF (7) can be extended to incorporate additional information:

- **Known features for time series:** In many applications, one is given additional features along with the observed time series. Specifically, given a set of feature vectors $\{a_i \in \mathbb{R}^d\}$ for each row of $Y$, we can look to solve

$$\min_{F,X,\Theta} \sum_{(i,t)\in\Omega} \left(Y_{it} - a_i^\top F x_t\right)^2 + \lambda_f \mathcal{R}_f(F)$$
$$+ \lambda_x \mathcal{T}_\mathrm{M}(X \mid \Theta) + \lambda_\theta \mathcal{R}_\theta(\Theta). \tag{15}$$

  That is, the observation $Y_{it}$ is posited to be a bilinear function of the feature vector $a_i$ and the latent vector $x_t$. Such an inductive framework has two advantages: we can generalize TRMF to a new time series without any observations up to time $T$ (i.e., a new row $i'$ of $Y$ without any observations). As long as the feature vector $a_{i'}$ is available, the model learned by TRMF can be used to estimate $Y_{i't} = a_{i'}^\top F x_t$, $\forall t$. Furthermore, prediction can be significantly sped up when $d \ll n$, since the dimension of $F$ is reduced from $n \times k$ to $d \times k$. Such methods for standard multi-label learning and matrix completion have been previously considered in [6, 24, 25].

- **Graph information among time series:** Often, separate features for the time series are not known, but other relational information is available. When a graph that encodes pairwise interactions among multiple time series is available, one can incorporate this graph in our framework using the graph regularization approach (2). Such cases are common in inventory and sales tracking, where sales of one item is related to sales of other items. Given a graph $G^f$ describing the relationship among multiple time series, we can formulate a graph regularized problem:

$$\min_{F,X,\Theta} \sum_{(i,t)\in\Omega} \left(Y_{it} - f_i^\top x_t\right)^2 + \lambda_f \mathcal{G}\left(F \mid G^f, \eta\right)$$
$$+ \lambda_x \mathcal{T}_\mathrm{M}(X \mid \Theta) + \lambda_\theta \mathcal{R}_\theta(\Theta), \tag{16}$$

  where $\mathcal{G}\left(F \mid G^f, \eta\right)$ is the graph regularizer defined in (2) capturing pairwise interactions between time series. Graph regularized matrix completion methods have been previously considered in [15, 27].

- **Temporal-regularized tensor factorization:** Naturally, TRMF can be easily extended to analyze temporal collaborative filtering applications [19, 23], where the targeted data is a tensor with certain modes evolving over time. For example, consider $\mathcal{Y} \in \mathbb{R}^{m \times n \times T}$ be a 3-way tensor with $Y_{ijt}$ encoding the rating of the $i$-th user for the $j$-th item at time point $t$. We can consider the following temporal regularization tensor factorization (TRTF) with $\mathcal{T}_\mathrm{M}(X \mid \Theta)$ as follows:

$$\min_{P,Q,X,\Theta} \sum_{(i,j,t)\in\Omega} \left(Y_{ijt} - \langle p_i, q_j, x_t \rangle\right)^2 + \lambda_p \mathcal{R}_p(P)$$
$$+ \mathcal{R}_q(Q) + \mathcal{T}_\mathrm{M}(X \mid \Theta) + \mathcal{R}_\theta(\Theta), \tag{17}$$

  where $P = [p_1, \cdots, p_m]^\top \in \mathbb{R}^{m \times k}$ and $Q = [q_1, \cdots, q_n]^\top \in \mathbb{R}^{n \times k}$ are the latent embeddings for the $m$ users and $n$ items, respectively, and with some abuse of notation, we define $\langle p_i, q_j, x_t \rangle = \sum_{r=1}^k p_{ir} q_{jr} x_{tr}$.

# C  Proofs

## C.1  Proof of Theorem 1

*Proof.* In this proof, we use the notations and summation manipulation techniques introduced by Knuth [4]. To prove (11), it suffices to prove that

$$\sum_{m \le t \le T} \left(\sum_{l \in \bar{\mathcal{L}}} w_l x_{t-l}\right)^2 = \sum_{1 \le t \le T} \sum_{1 \le d \le L} G^\mathrm{AR}_{t,t+d}(x_t - x_{t+d})^2 + \bar{x}^\top D \bar{x}. \tag{18}$$

The LHS of the (18) can be expanded and regrouped as follows.

$$\sum_{m \leq t \leq T} \left( \sum_{l \in \bar{\mathcal{L}}} w_l x_{t-l} \right)^2$$

$$= \sum_{m \leq t \leq T} \left( \sum_{l \in \bar{\mathcal{L}}} w_l^2 x_{t-l}^2 + \sum_{1 \leq d \leq L} \sum_{l \in \delta(d)} 2 w_l w_{l-d} x_{t-l} x_{t-l+d} \right)$$

$$= \sum_{m \leq t \leq T} \left( \sum_{l \in \bar{\mathcal{L}}} w_l^2 x_{t-l}^2 + \sum_{1 \leq d \leq L} \sum_{l \in \delta(d)} \left( -w_l w_{l-d} (x_{t-l} - x_{t-l+d})^2 + w_l w_{l-d} (x_{t-l}^2 + x_{t-l+d}^2) \right) \right)$$

$$= \underbrace{\sum_{m \leq t \leq T} \sum_{1 \leq d \leq L} \sum_{l \in \delta(d)} -w_l w_{l-d} (x_{t-l} - x_{t-l+d})^2}_{\mathcal{G}(\bar{x})} + \underbrace{\sum_{m \leq t \leq T} \left( \sum_{l \in \bar{\mathcal{L}}} w_l^2 x_{t-l}^2 + \sum_{1 \leq d \leq L} \sum_{l \in \delta(d)} w_l w_{l-d} (x_{t-l}^2 + x_{t-l+d}^2) \right)}_{\mathcal{D}(\bar{x})}$$

Let's look at the first term $\mathcal{G}(\bar{x})$:

$$\mathcal{G}(\bar{x}) = \sum_{1 \leq d \leq L} \sum_{l \in \delta(d)} \sum_{m \leq t \leq T} -w_l w_{l-d} (x_{t-l} - x_{t-l+d})^2$$

$$= \sum_{1 \leq d \leq L} \sum_{l \in \delta(d)} \sum_{m-l \leq t \leq T-l} -w_l w_{l-d} (x_t - x_{t+d})^2$$

$$= \sum_{1 \leq d \leq L} \sum_{l \in \delta(d)} \sum_{1 \leq t \leq T} -w_l w_{l-d} (x_t - x_{t+d})^2 [m-l \leq t \leq T-l]$$

$$= \sum_{1 \leq t \leq T} \sum_{1 \leq d \leq L} \left( \sum_{l \in \delta(d)} -w_l w_{l-d} [m-l \leq t \leq T-l] \right) (x_t - x_{t+d})^2$$

$$= \sum_{1 \leq t \leq T} \sum_{1 \leq d \leq L} \underbrace{\left( \sum_{\substack{l \in \delta(d) \\ m \leq t+l \leq T}} -w_l w_{l-d} \right)}_{G_{t,t+d}} (x_t - x_{t+d})^2,$$

where we can see that $\mathcal{G}(\bar{x})$ is equivalent to the first term of RHS of (18).

Now, we consider the second term $\mathcal{D}(\bar{x})$:

$$\mathcal{D}(\bar{x}) = \sum_{m \leq t \leq T} \left( \sum_{l \in \bar{\mathcal{L}}} w_l^2 x_{t-l}^2 + \sum_{1 \leq d \leq L} \sum_{l \in \delta(d)} w_l w_{l-d} (x_{t-l}^2 + x_{t-l+d}^2) \right)$$

$$= \underbrace{\sum_{m \leq t \leq T} \sum_{l \in \bar{\mathcal{L}}} w_l^2 x_{t-l}^2}_{\mathcal{D}_1(\bar{x})} + \underbrace{\sum_{m \leq t \leq T} \sum_{1 \leq d \leq L} \sum_{l \in \delta(d)} w_l w_{l-d} x_{t-l}^2}_{\mathcal{D}_2(\bar{x})} + \underbrace{\sum_{m \leq t \leq T} \sum_{1 \leq d \leq L} \sum_{l \in \delta(d)} w_l w_{l-d} x_{t-l+d}^2}_{\mathcal{D}_3(\bar{x})}$$

$$\mathcal{D}_1(\bar{\boldsymbol{x}}) = \sum_{l \in \bar{\mathcal{L}}} \sum_{m \leq t \leq T} w_l^2 x_{t-l}^2 = \sum_{l \in \bar{\mathcal{L}}} \sum_{m-l \leq t \leq T-l} w_l^2 x_t^2 = \sum_{1 \leq t \leq T} \left( \sum_{l \in \bar{\mathcal{L}}} w_l^2 [m \leq t + l \leq T] \right) x_t^2$$

$$= \sum_{1 \leq t \leq T} \left( \sum_{l,l' \in \bar{\mathcal{L}}} w_l w_{l'} [m \leq t + l \leq T][l' = l] \right) x_t^2$$

$$\mathcal{D}_2(\bar{\boldsymbol{x}}) = \sum_{m \leq t \leq T} \sum_{1 \leq d \leq L} \sum_{l \in \delta(d)} w_l w_{l-d} x_{t-l}^2 = \sum_{1 \leq t \leq T} \left( \sum_{1 \leq d \leq L} \sum_{l \in \delta(d)} w_l w_{l-d} [m \leq t + l \leq T] \right) x_t^2$$

$$= \sum_{1 \leq t \leq T} \left( \sum_{l,l' \in \bar{\mathcal{L}}} w_l w_{l'} [m \leq t + l \leq T][l' < l] \right) x_t^2$$

$$\mathcal{D}_3(\bar{\boldsymbol{x}}) = \sum_{m \leq t \leq T} \sum_{1 \leq d \leq L} \sum_{l \in \delta(d)} w_l w_{l-d} x_{t-l+d}^2 = \sum_{1 \leq t \leq T} \left( \sum_{1 \leq d \leq L} \sum_{l \in \delta(d)} w_l w_{l-d} [m \leq t + l - d \leq T] \right) x_t^2$$

$$= \sum_{1 \leq t \leq T} \left( \sum_{l',l \in \bar{\mathcal{L}}} w_l w_{l'} [m \leq t + l \leq T][l' > l] \right) x_t^2$$

Let $D \in R^{T \times T}$ be a diagonal matrix with $D_{tt}$ be the coefficient associated with $x_t^2$ in $\mathcal{D}(\bar{\boldsymbol{x}})$. Combining the results of $\mathcal{D}_1(\bar{\boldsymbol{x}}), \mathcal{D}_2(\bar{\boldsymbol{x}})$, and $\mathcal{D}_3(\bar{\boldsymbol{x}})$, $D_t$ can be written as follows.

$$D_{tt} = \left( \sum_{l \in \bar{\mathcal{L}}} w_l \right) \left( \sum_{l \in \bar{\mathcal{L}}} w_l [m \leq t + l \leq T] \right) \quad \forall t.$$

It is clear that $\mathcal{D}(\bar{\boldsymbol{x}}) = \bar{\boldsymbol{x}}^\top D \bar{\boldsymbol{x}}$. Note that $\forall t = m, \ldots, T - L$, $D_{tt} = \left( \sum_{l \in \bar{\mathcal{L}}} w_l \right)^2$. $\qquad \square$

### C.2 Proof of Corollary 1

*Proof.* It is well known that graph regularization can be written in the quadratic form [18] as follows.

$$\frac{1}{2} \sum_{t \sim s} G_{ts} (x_t - x_s)^2 = \bar{\boldsymbol{x}}^\top \mathbf{Lap}(G) \bar{\boldsymbol{x}},$$

where $\mathbf{Lap}(G)$ is the $T \times T$ graph Laplacian for $G$ defined as:

$$\mathbf{Lap}(G)_{ts} = \begin{cases} \sum_j G_{tj}, & t = s \\ -G_{ts}, & t \neq s \text{ and there is an edge } t \sim s \\ 0, & \text{otherwise.} \end{cases}$$

Based on the above fact and the results from Theorem 1, we obtain the quadratic form for $\mathcal{T}_{\text{AR}}(\bar{\boldsymbol{x}} | \mathcal{L}, \bar{\boldsymbol{w}}, \eta)$ as follows.

$$\mathcal{T}_{\text{AR}}(\bar{\boldsymbol{x}} | \mathcal{L}, \bar{\boldsymbol{w}}, \eta) = \frac{1}{2} \bar{\boldsymbol{x}}^\top \left( \mathbf{Lap}(G^{\text{AR}}) + \underbrace{D + \eta I}_{\text{diagonal}} \right) \bar{\boldsymbol{x}}.$$

Because $D + \eta I$ is diagonal, the non-zero pattern of the off-diagonal entries of the inverse covariance $\Sigma_{\text{AR}}^{-1}$ for $\mathcal{T}_{\text{AR}}(\bar{\boldsymbol{x}} | \mathcal{L}, \bar{\boldsymbol{w}}, \eta)$ is determined by $\mathbf{Lap}(G^{\text{AR}})$ which shares the same non-zero pattern as $G^{\text{AR}}$. $\qquad \square$

### C.3 Details for Corollary 2

We use results developed in [15] to arrive at our result. Assume we are given the matrix $B$. We first define the following quantities:

$$\alpha := \sqrt{nT} \frac{\|Z^\star B\|_\infty}{\|Z^\star B\|_F} \qquad \beta := \frac{\|Z^\star B\|_*}{\|Z^\star B\|_F} \tag{19}$$

where the $\|\cdot\|_\infty$ norm is taken element-wise. Note that, the above quantities capture the "simplicity" of the matrix $Z$. For example, a small value of $\alpha$ implies that the matrix $ZB$ is not overly spiky,

meaning that the entries of the matrix are well spread out in magnitude. Next, define the set

$$\mathcal{C} := \left\{ Z : \alpha\beta \leq C\sqrt{\frac{N}{\log(n+T)}} \right\}, \tag{20}$$

with $C$ being a constant that depends on $\alpha$.

Finally, we assume the following observation model: for each $i, j \in \Omega$, suppose we observe

$$Y_{ij} = Z_{ij}^{\star} + \frac{\sigma}{\sqrt{nT}}\eta_{ij} \quad \eta_{ij} \sim \mathcal{N}(0,1)$$

Then, we can see that the setup is identical to that considered in [15] with the difference being that there is no graph present that relates the rows of $Z^{\star}$. Hence, setting $A = I$ in Theorem 1 in the aforementioned paper yields our result.

## D Details: Scalability Issue of R-DLM package

In this section, we show the source code demonstrating that R-DLM fails to handle high-dimensional time series even with $n = 32$. Interested readers can run the following R code to see that the `dlmMLE()` function in R-DLM is able to run on a 16-dimensional time series. However, when we increase the dimension to 32, `dlmMLE()` crashes the entire R program.

```
library(dlm)
builderFactory <- function(n,k) {
    n = n;
    k = k;
    init = c(rep(0,k), rep(0.1,3),0.1*rnorm(n*k), 0.1*rnorm(k*k))
    build = function(x) {
        m0 = x[1:k]
        C0 = (abs(x[k+1]))*diag(k)
        V  = (abs(x[k+2]))*diag(n)
        W  = (abs(x[k+3]))*diag(k)
        FF = matrix(nrow=n,ncol=k, data=x[(k+3+1):(k+3+n*k)])
        GG = matrix(nrow=k,ncol=k, data=x[(k+3+n*k+1):(k+3+n*k+k*k)])
        return (dlm( m0=m0, C0=C0, FF=FF, GG=GG, V=V, W=W))
    }
    return (list(n=n,k=k,init=init,build=build))
}

Rdlm_train <- function(Y, k, maxit) {
    if(missing(maxit)) { maxit=10 }

    if(ncol(Y)==3) {
        Ymat = matrix(nrow=max(Y(,1)),ncol=max(Y(,2)))
        Ymat[cbind(Y(,1),Y(,2))] = Y(,3)
    } else {
        Ymat = Y;
    }
    n = nrow(Ymat)
    TT = ncol(Ymat)
    dlm_builder = builderFactory(n, k)
    mle = dlmMLE(Ymat,dlm_builder$init,build=dlm_builder$build,
                control=list(maxit=10))
    dlm = dlm_builder$build(mle$par)
    dlm_filt = dlmFilter(Ymat,dlm)
    return (dlm_filt)
}

tmp = t(as.matrix(Nile));
tmp=rbind(tmp,tmp); tmp=rbind(tmp,tmp);
tmp=rbind(tmp,tmp); tmp=rbind(tmp,tmp);

print(nrow(tmp))
```

```
Rdlm_train(tmp,4);
print('works')

tmp=rbind(tmp,tmp);
print(nrow(tmp))
Rdlm_train(tmp,4);
```

## Footnotes

[3]https://archive.ics.uci.edu/ml/datasets/ElectricityLoadDiagrams20112014.

[4]https://archive.ics.uci.edu/ml/datasets/PEMS-SF.