[Reviews · NeurIPS 2016]

Reviewer 1

Summary

The paper addresses the much relevant problem of time series forecasting through matrix factorization. The approach factorizes a matrix where the rows are the time series and the columns represent the time dimension. Forecasting is then reduced to predicting the "missing" rows of the matrix , which cannot be trivially done, since the latent features for the missing columns cannot be learned. The authors propose to address this issue by adding an auto-regression term for predicting the missing latent features. This term is added as a regularizer on the loss function. Experiments on synthetic and real data show a substantial improvement over the state-of-the-art.

Qualitative Assessment

This is a very nice well written paper, which addresses a challenging problem with a principled solution. The experiments are very convincing.

Confidence in this Review

2-Confident (read it all; understood it all reasonably well)


Reviewer 2

Summary

The paper presents a temporal regularized matrix factorization algorithm to solve large-scale multidimensional time series prediction problem efficiently. The authors claim that the proposed algorithm is two orders of magnitude faster than the competing algorithms on some real-world datasets.

Qualitative Assessment

Two popularly used time series prediction models, autoregressive (AR) and dynamic linear models (DLM), are both time consuming to learn, especially for high dimensional time series prediction problem with missing values. However, matrix factorization is relatively efficient for large-scale matrix. The authors model the high dimensional time series as matrix and induce the constraints as regularization terms, then formulate the problem as a regularized matrix factorization problem and solve it by adopting the off-the-shelf solvers. The temporal regularized matrix factorization(TRMF) framework proposed by the paper sounds interesting. Inherited from the properties of MF, TRMF is able to deal with missing values and can be scalable to high-dimensional time series datasets. The regularizer proposed in the paper is able to incorporate the temporal dependency, hence the model has better forecast ability comparing to traditionaly time series model. The graph regularization can incorporate temporal dependencies but unable to support negatively correlated dependencies. TRMF is able to support negative correlations and the weights can be learned automatically from datasets, which makes TRMF have better forecasting ability and missing value imputation ability than other methods. Overall the paper is clear written. I have few questions as follows. 1. The authors claim that the AR model is time consuming to learn even in one-dimensional case. However, the proposed algorithm use decoupled multiple one dimensional AR model as a subproblem. 2. Does the sparsity of the matrix (missing ratio) matter to the performance improvement of TRMP over AR or DLM? 3. Does this sparsity matter to the efficiency of the algorithm? 4. Is there any assumption that makes TRMF work? In practical problem, especially in multi-dimensional time series prediction, the multiple time series usually have very different behaviors, while matrix factorization problem often assume that the matrix has very low rank. Could the authors provide some analysis on this issue?

Confidence in this Review

2-Confident (read it all; understood it all reasonably well)


Reviewer 3

Summary

The paper presents a temporal regularized matrix factorization (TRMF) framework that supports data-driven temporal learning and forecasting and shows the connections to many existing approaches for time series analysis. It evaluate the performance and shows the improved performance of TRMF in terms of scalability and prediction quality.

Qualitative Assessment

The paper in general is pleasant to read and written well. The problem is motivated and the article is well structured. It does a good job on showing the connections to prior work and the limitations of prior work. The idea of using time series model for regularization is also interesting. The experimental results also show the improvement with this regularization.

Confidence in this Review

1-Less confident (might not have understood significant parts)


Reviewer 4

Summary

The authors offer a framework that generalizes previous Temporal Matrix Factorization approaches that extends graph-based regularization approaches to support negative edge weights (negative correlation between points in time). This allows TRMF to scale (as compared to classic AR or DLM models) as well as to infer the temporal structure during the learning task. This framework was then applied to an autoregressive structure and demonstrated superior results to some state-of-the-art baselines.

Qualitative Assessment

This writing was fluid and the ideas were presented in the right succession and with adequate motivation. The idea seems rather novel even if it's an extension of graph-based regularization for Matrix Factorization. An explanation of why the basic graph-based regularization does not support negative edge weight would have been appreciated. Interesting connection to temporal collaborative filtering but there have been recent development in dynamic recommender systems / collaborative filtering that do not necessarily force an identity weight matrix (as claimed in line 257). The results looks quite impressive compared to the selected baselines, especially for value imputation. However, it would have been interesting to compare to more recent dynamic factorization models or even ones older than TCF for both the forecasting and the imputation. Finally, Wilson et al. had what looks like a similar approach (but not exactly and definitely not derived the same way) in "Regularized non-negative matrix factorization with temporal dependencies for speech denoising." A comparison of the works/results would've been appreciated since this other work claimed state-of-the-art results at the time (2008).

Confidence in this Review

2-Confident (read it all; understood it all reasonably well)


Reviewer 5

Summary

Driven by the characteristics of high-dimensional and noise in real world time series data, this paper aims to propose a scalable and robust framework to tackle the challenges. Specifically, the paper presents a novel Temporal Regularized Matrix Factorization (TRMF) model, which incorporate a temporal regularization module to traditional Matrix Factorization method. In addition, various existing methods are carefully compared to illustrate the novelty of the proposed framework. Moreover, this framework is specifically applied to traditional Autoregressive (AR) model and shows the ability to model both positive and negative dependency correlations and flexibility for choosing lag length. Finally, experiments on synthetic and real world data set shows the effectiveness of aforementioned strengths on TRMF framework. In a nutshell, this paper presents a novel framework to solve an interesting problem motivated by real world time series applications. The mathematic induction and proof is solid and experiments are well conducted.

Qualitative Assessment

Strengths Compared with previous methods for time series prediction, TRMF is able to: 1) Capture the temporal dependency of time series data instance via regularization term; 2) Impute missing values to handle data noise and 3) Scale to large dimension of time series data. So this framework could have potential to be very useful with high impact. Weaknesses 1. The time complexity of the learning algorithm should be explicitly estimated to proof the scalability properties. 2. In Figure 4, the time complexity for TRMF-AR({1,8}) and TRMF-AR({1,2,…,8}) seems to be the same. The reason should be explained.

Confidence in this Review

2-Confident (read it all; understood it all reasonably well)


Reviewer 6

Summary

This paper presents an approach to improve matrix factorization framework in the context of forecasting, where one of the matrix dimensions is time. The main contribution is the proposal of a regularization term that is the log-likelihood of a temporal model regarding some parameters to estimate. In fact, this is a funny way to put it. What their model is really doing is generating n time series by linearly combining k latent time series, governed by a Gauss-Markov model, and where the n weights of the linear combinations can be learned. This approach has several benefits, first it can handle missing-value easily, second it can share information across different time-series while keeping temporal coherency in the model. Contrarily to graph-based regularization, the weights of edges do not have to be specified but can be learned and negative correlation are supported. The author instantiate this framework with a simple AR model. In this special case, they show how the temporal regularizer can be expressed as graph regularization which allow to use common solvers. Finally, they do comparison on 5 datasets with improvement in running time and around 10% improvement in accuracy compared to other methods.

Qualitative Assessment

The paper is well written (except maybe I find the formulation of the time series part as "regularizer" overly complex, see below), and also well motivated. The proposed method does better than a range of competitors in experiments on a set of datasets, some of which are proprietary (which is fine with me). Finally, I find the application domain (retail forecasting) exciting and of high impact, and would like to see more work in the NIPS community in this direction (even though the lack of publicly available forecasting datasets of sufficient quality is clearly a hindrance). On the other hand, I am somewhat disappointed by the overall scope of the work. In fact, what is proposed here is to model a smaller set of k dependent time series in a standard fashion (correlated AR), and then map these to n (partially) observed series using n linear weight vectors. That is all. Phrasing this as "regularization" is uncommon and does not help the understanding. I am not an expert on work in this area, but find it hard to accept that something like this has not been attempted before. There is no mentioning of the fact that demand forecasting distributions are typically intermittent, and therefore poorly modelled by Gaussian noise (I recall an ICML paper by Chapados a few years ago that made this point). Also, there is no mentioning of basic structural elements, such as seasonality, promotions, or other features. For example, Walmart will sell more just before Christmas, and more turkey before Thanksgiving. In fact, what is described here, does not depend on other data than the demand targets at all. I doubt that a method like this would work very well on real demand data. The second problem is the range of methods the authors compare against. Given that the proposed method is not very realistic as far as forecasting goes (no seasonality, no external inputs such as promotions, ...), it is understandable the authors compare against equally simple methods. But why not compare against simple things like Hyndman's forecast package, run separately on each series? This would take seasonality factors into account. The authors also do not try other basic setups in the ARIMA space. In fact, I am not convinced from these results that MF would in fact outperform an approach of individually forecasting each series, if only this was done with best practices. I have the strong feeling that a paper with this kind of evaluation would not pass at a forecasting conference or journal. It also seems a bit unfair in my opinion to use only AR(1) in the comparison whereas domain knowledge was used to specify the lags of TRMF-AR. From the running time mentioned in Figure 5, at least AR({1,24}) for hourly grain and AR({1,52}) for weekly grain experiments can be used. The scalability argument given by the author versus AR is also contrasted by the fact that AR is embarrassingly parallel and can be trained in multiple machine without difficulties allowing to scale to arbitrary large number whereas the proposed method does not enjoy this property. Detailed comments: p5: “unlike most approaches mentioned in section 2.2”. This sounds a bit vague to me, probably this refers only to Graph based approach. However in the second point, it says that the methods cant handle different lag which is not true for Graph based methods. p6: please add Coefficient of Variation in the table to account for the forecast difficulty in Table 1. p6: About Walmart dataset, the missing ratios indicated in Table 1 are not the initial one. The appendix mentions that time-series are aligned by adding unobserved data points. Please also add the proportion of unobserved data in the main document of the initial time-series, the proportion is probably much smaller and this gives the wrong impression if the reader does not check the appendix (if this ratio is very small then all methods could be tested in table 2 by replacing unobserved by zeros). p7: although it can be understood in next page with the lag chosen for electricity and traffic datasets, it will help to specify the hourly time grain sooner, it is only stated explicitly in the appendix. p7: Figure 4. the values of n (500, 1000, 5000, 10000, 50000) are not easy to read. W p8: R-DLM should could be removed as it cannot handle any of the datasets except the articial one (maybe just mention that it only handles n lower than 32 in the text).

Confidence in this Review

2-Confident (read it all; understood it all reasonably well)